# Balancing tourism development and habitat conservation in fragile ecosystems: A case study of the Qinghai-Tibet Plateau

Bo Yang[1,2]*, Jiawei Wu[1,2], Anna Miao[1], Jinlu Ran[1], Rong Jia[1]

1 College of Tourism, Northwest Normal University, Lanzhou, China, 2 Hexi Corridor Research Institute, Northwest Normal University, Lanzhou, China

* yangbo907@126.com

## Abstract

As a typical representative of global ecologically fragile areas and emerging tourism hotspots, the Qinghai-Tibet Plateau has important research value in its collaborative path between ecosystem protection and tourism development. We take Gannan Tibetan Autonomous Prefecture (GTAP) in the northeast of the Qinghai-Tibet Plateau as a case, and use InVEST model, kernel density estimation and geodetector methods to reveal the spatial distribution of habitat quality from multiple scales, quantify the intensity of tourism activities, and explore the impact of tourism activities on fragile ecosystems. The results show that: (1) Habitat quality shows significant gradient characteristics, and the overall decreases from southeast to northwest. At the county scale, Diebu County in the southeast (0.82) and Maqu County in the northwest (0.31) form a polar contrast; the differences in township scales are more significant, and Chagang Township in the Zhouqu County (0.89) and Yuzhong Street in the Hezuo city (0.18) respectively represent the optimal and worst habitat units. (2) Tourism development presents a "core-transition- marginal" circle structure, Xiahe, Hezuo and other northern counties and cities to form the core of the development of factor concentration (kernel density value > 3.5), Luqu County for the transition zone, Maqu County is in the development of the marginal area. (3) Analysis of geodetector shows that topographic factors (elevation $q = 0.62$, slope $q = 0.58$) dominate the natural background distinction, while tourism factors ($q = 0.71$) become the primary man-made driving force in the core development area. It is worth noting that the interaction between nature and man-made elements shows a nonlinear enhancement effect (interaction $q$ value > 0.85). (4) The intensity of tourism activities in GTAP is negatively correlated with habitat quality. Tourism activities are an important artificial driving force for the different spatial distribution of habitat quality in the tourism areas of the Qinghai-Tibet Plateau. This case study reveals that the differences in spatial development caused by tourism fever on the Qinghai-Tibet Plateau are reshaping the ecosystem pattern. This study proposes a two-dimensional assessment framework

**Data availability statement:** All relevant data are within the manuscript and its Supporting Information files.

**Funding:** This research was supported by the 2021 Young Teachers' Scientific Research Ability Improvement Plan of Northwest Normal University, the Innovation Fund for Colleges and Universities of the Department of Education, Gansu Province, China (grant no. 2021B-079), 2025 Graduate Research "Scientific Research Funding" Project, Gansu Province, China, and 2025 Graduate Research "Innovation Star" Project, Gansu Province, China.

**Competing interests:** The authors have declared that no competing interests exist.

of "ecological sensitivity-tourism pressure", which provides scientific support for the development of differentiated ecotourism management strategies. Recommends that dynamic monitoring of ecological carrying capacity be implemented in the core area of the development, strengthen natural restoration and tourism relocation in the marginal areas, and explore ecological tourism models with community participation in the transition zone. This paper has important practical value for the synergistic promotion of biodiversity conservation and sustainable development of tourism in the plateau region of the Tibetan Plateau.

## 1. Introduction

According to the United Nations "World Population Outlook (2024)", the global population will exceed 10.3 billion by the mid-2080s, an increase of 25.6% from 2024 [1]. The expansion of the global population base and the increase in mobility directly stimulate the large-scale expansion of the tourism industry. The tourism industry has transformed from "economic growth with biological" to a new driving force for reshaping the relationship between people and land [2]. The prosperity of tourism can often promote the economic and cultural development of tourist destinations, such as increasing economic income, developing cultural education and stimulating a wide range of infrastructure construction (including tourist attractions, transportation hubs and resorts, etc.). However, the development of tourism is often accompanied by changes in land use, which is one of the most important driving forces for ecosystem services [3,4]. Non-natural factors of land use change can, to varying degrees, have far-reaching impacts on environmental quality [5], water cycle [6], carbon emissions [7], and biodiversity [8] in the region. Ecologically fragile areas are often economically underdeveloped regions, and the impact of tourism activities on land use change is most prominent in ecologically fragile areas [9]. Managers usually attach importance to the economic benefits brought by tourism development and ignore its impact on the environment, resulting in great damage to the ecosystem after tourism development exceeds the threshold of natural recovery [10]. Therefore, how to balance the relationship between tourism development and ecological protection in fragile ecosystems in the ecologically fragile areas represented by the Tibetan Plateau is an issue that managers, planners and tourism developers should continue to explore. The relationship between these two is of great significance in realizing the harmonization of tourism development and ecological protection in ecologically fragile areas.

Habitat quality (HQ) is the ability of the natural environment to provide suitable conditions for the continued survival and development of individuals and populations, it is a reflection of the size of the natural environment's ability to support the survival and reproduction of organisms, and a key indicator of ecosystem service function and biodiversity level [11]. The quality of habitat not only reflects the ecosystem service capacity and biodiversity level, but also directly affects ecological security and human well-being [12,13]. In recent years, with the rapid development of computer technology, 3S (GPS, GIS, RS) technology and big data technology, a series of ecological

models have been introduced for quantitative assessment of habitat quality, including the ecological niche model (ENM) [14], the habitat suitability index model (HSIM) [15], the social value model of ecosystem services (SolVES) [16], the multi-scale ecosystem service integration model (MIMES) [17], the artificial intelligence of ecosystem service model (ARIES) [18], the pressure-state-response model (PSR) [19], and the Integrated Valuation of Ecosystem Services and Tradeoffs model(InVEST) [20–22]. Among them, the InVEST model is widely used because of its ease of application, data availability, and strong theoretical foundation, which makes it possible to conduct studies at different scales [23,24].

In related research on habitat quality, research on habitat quality assessment [25], analysis of spatiotemporal evolution characteristics [26–28], driving factors [29] and influencing mechanism [30] have become key issues. For example, Parvar assessed spatial and temporal changes in HQ in the Iranian Plateau using the optimized InVEST model, and their findings highlighted that long-term climate change and urbanization processes have led to a continuous decline in HQ over the past 30 years, and that HQ is predicted to decrease further by 2050 [31]; Yohannes analyzed the main causes of degradation of HQ in the Ethiopian highlands from 1972–2017, noting that landscape composition was greater than watershed impacts [32]; Han used mathematical statistics combined with InVEST model to reveal the impact mechanism of HQ in the arid urban agglomeration in the ecologically fragile areas of the northern slope of Tianshan Mountain, explored the interaction between natural and unnatural factors, and established an ecological-economic influence path [33]. In recent years, the impact of tourism-based human activities on biodiversity and HQ in ecologically fragile areas has become a hot topic of research. Luo et al. discussed the impact of different tourism interference intensity on the habitat quality and population size of giant salamanders in the Zhangjiajie Nature Reserve in Hunan Province, and found that noise and pathogens brought by tourism can directly or indirectly reduce the size of giant salamanders populations [34]; Taking Yulong River Basin as an example, Sun et al. explored the coupling relationship between "tourism development-land use-landscape pattern" in ecologically fragile karst areas based on 3S technology and POI spatial analysis [35], it provides a new reference for the impact of the rapid development of tourism on land use change and ecological environment; Hu et al. used the InVEST habitat quality and KDE method, combined with the distribution density of ethnic minority village spaces, to determine the potential development space of QMDA tourism resources, emphasizing the delicate balance between tourism development and ecosystem protection [36]. However, relevant research on HQ in ecologically fragile zones mainly stays from the perspective of ecology and geography, and there seems to be very few research on exploring the connection between tourism development and ecological protection.

The Tibetan Plateau region, with its unique geomorphologic and climatic conditions, has become an ecological protection barrier in China and at the same time a typical ecologically fragile area [37]. In recent years, it has not only become a key area for HQ research, but also one of the most popular tourist destinations in China due to its unique ecological and cultural values [38,39]. In 2024, the Tibetan Plateau region will receive 117 million tourists and realize a comprehensive tourism income of 126.659 billion yuan. As a typical interactive area of human-land relationship, the tourism land on the Tibetan Plateau combines high-quality natural and humanistic resources, but at the same time there is a contradiction between ecological protection and tourism development [40]. Currently, most of the research on tourism sites on the Tibetan Plateau focuses on exploring the development of tourism industry, tourism safety assessment, and tourism competitiveness enhancement. In contrast, there is still relatively little exploration of its HQ, and the relationship between its HQ and tourism development needs to be further clarified. Therefore, we selected GTAP, a typical plateau tourist at the northeast edge of the Qinghai-Tibet Plateau, and deeply explored the spatial distribution characteristics and main influencing factors of its habitat quality, revealing the impact of tourism development on habitat quality in the Qinghai-Tibet Plateau area. The area has a unique plateau ecosystem and rich Tibetan cultural tourism resources. The rapid development of tourism has become one of the important forms of human activities. As of 2023, GTAP has a total of 36 A-level scenic spots, receiving 22 million tourists throughout the year and achieving tourism revenue of 11 billion yuan. The tourism industry has become an important pillar of economic development. However, the rapid development of tourism activities and related infrastructure construction has exacerbated grassland degradation, forest reduction and wetland shrinkage, posing a threat to habitat quality and ecosystem stability [41].

For this reason, it is urgent to conduct an in-depth study on the impact of tourism development on Gannan's habitat quality, and to clarify the spatial differentiation and influencing factors of Gannan's habitat quality, which has become a key task to promote the sustainable development of Gannan's tourism. We use the InVEST model as an evaluation tool, and combines the methods of KDE and geodetector to explore the relationship between tourism development and ecological protection in GTAP, a typical tourist destination in the northeastern margin of the Qinghai-Tibet Plateau. Specifically, we aim to: (1) Reveal the spatial distribution characteristics of HQs at different scales in GTAP; (2) Reveal the distribution density of tourism facilities and divide the intensity zone of tourism activity; (3) Explore the driving factors of habitat quality differences under different tourist activity intensity zoning. (4) Explore the relationship between differences in habitat quality and intensity of tourism activities and develop targeted recommendations.

## 2. Materials and methods

### 2.1 Study area

Gannan Tibetan Autonomous Prefecture (GTAP) of Gansu Province (Fig 1), is located on the northeast edge of the Qinghai-Tibet Plateau (33°06' – 35°10' N, 100°45' – 104°35' E), it covers various ecosystems such as grasslands, forests, rivers, etc. It is located in the upper reaches of the Yellow River and has an extremely sensitive ecological environment. Governed by 7 counties and 1 city, with a total area of about 36000 km², GTAP has a complex and diverse topography, an important ecological status, and is a water conservation area and replenishment area of the Yellow River and the Yangtze River, with rich natural resources and unique humanistic landscapes. The terrain is generally characterized by high in the southwest and low in the northeast, with the Animaqing Mountains and Xiqin Mountains in the southwest, the Minshan Mountains and Dieshan Mountains in the southeast, and the Grassland Area in the north. At the end of 2023, the total output value of GTAP was RMB 26.081 billion, with a resident population of 668700, concentrated in the river valley areas of Cooperation City and Lintan County in the north and Zhouqu County in the south. As the "window to the Tibetan Plateau" and the "springboard for Tibetan modernization", GTAP has been listed as an ecological main function area and an ecological civilization advance demonstration area by the state. As a result of long-term human activities and climate change, GTAP faces ecological problems such as grassland degradation and forest decline, and is in urgent need of enhanced ecological environment assessment and protection.

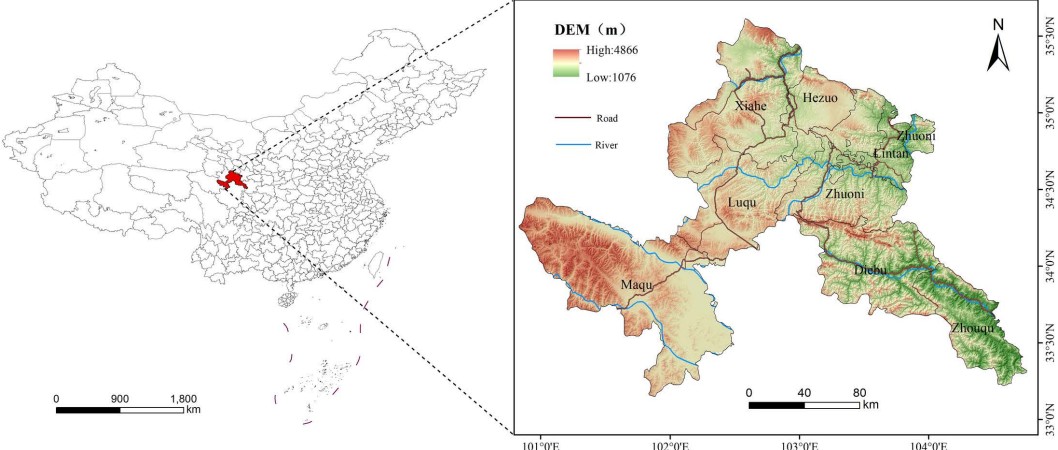

**Fig 1. Location of study area (base map and data from NGIC) Reprinted from NGIC under CC BY 4.0 license, with permission from "National Tibetan Plateau/ Third Pole Environment Data Center", original copyright 2017–2021.**

## 2.2. Data sources and processing

The data used in this article comes from multiple sources: (1) Land use remote sensing monitoring data for the 2020 and 2023 periods of GTAP (spatial resolution 30×30m), downloaded from the Resources and Environmental Science Data Center of the Chinese Academy of Sciences (http://www.resdc.cn/DataList.aspx), according to the national remote sensing monitoring land use/cover classification system, the first-level category includes six categories: forest land, grassland, cultivated land, wetland, construction land and unused land. The second-level category is further divided into 25 types based on the first-level category. (2) Digital elevation model (DEM) data with a spatial resolution of 30×30 m were downloaded from the Geospatial Data Cloud (http://www.gscloud.cn/), and slope and aspect data were extracted and generated from the digital elevation model (DEM) data. (3) The road network vector data comes from OpenStreetMap (OSM) (https://www.openstreetmap.org/), OSM is a free, open-source, and editable map service that provides users with accurate road spatial data. (4) GTAP administrative boundary vector data were obtained from National Tibetan Plateau/ Third Pole Environment Data Center (http://data.tpdc.ac.cn). (5) The POI data of 1826 restaurants, 3776 shopping stores, 118 tourist attractions, 558 leisure and entertainment spots and 1132 accommodation facilities in GTAP in 2023 were collected from AMap, Compare the number of accommodation facilities, hotels, tourist attractions, and transportation network area with the regional area to obtain data on hotel density, catering enterprise density, tourist attraction density, and transportation network density. (6) Data on population size, tourist trips, annual average temperature and annual cumulative precipitation data, and GDP (gross domestic product) in each district and county of GTAP were obtained from the 2024 Gannan Statistical Yearbook and the statistical bulletins of the districts and counties, and the regional population density was obtained by comparing the population size with the area of the region.

## 2.3. Methodology

### 2.3.1 Habitat quality assessment.

The InVEST-Habitat Quality modeling framework, jointly created through collaboration between Stanford University, TNC (The Nature Conservancy), and WWF (World Wide Fund For Nature), has gained broad acceptance in multi-scale habitat assessments [24,42] owing to its computational efficiency, enhanced geospatial explicitation capabilities, and interoperability with heterogeneous data sources. Our methodological framework implements the InVEST Habitat Quality module to conduct quantitative spatial analysis of habitat integrity and anthropogenic degradation patterns across GTAP during 2023. The habitat integrity quantification algorithm integrates four parametric dimensions: threat intensity weighting ($r_y$), land cover sensitivity coefficients ($S_{jr}$), threat decay functions based on Euclidean distances ($d_{jr}$), and regulatory protection modifiers ($\beta_x$). The habitat degradation index ($D_{xj}$) is derived through spatial convolution of threat factors using the equation:

$$D_{xj} = \sum_{r=1}^{R} \sum_{y=1}^{Y_r} \left( \frac{w_r}{\sum_{r=1}^{R} w_r} \right) r_y i_{rxy} \beta_x S_{jr} \tag{1}$$

$$i_{rxy} = 1 - \left( \frac{d_{xy}}{d_{r\max}} \right) \quad \text{(if linear)} \tag{2}$$

$$i_{rxy} = \exp \left( -\frac{2.99 d_{xy}}{d_{r\max}} \right) \quad \text{(if exponential)} \tag{3}$$

Formula parameters are defined as: $D_{xj}$ (Habitat Degradation Index) measures cumulative anthropogenic stress, where $R$ is total threat categories, $w_r$ is threat-specific weighting factors, $Y_r$ is spatial distribution grids, $r_y$ is normalized threat intensity per grid. Distance-decay effects are modeled $i_{rxy}$, where $\beta_x$ represents regulatory protection status (0 is protected, 1 is

unprotected). $S_{jr}$ quantifies habitat vulnerability coefficients, $d_{xy}$ is straight-line grid separation, $d_{rmax}$ is maximum effective radius of threat propagation. Elevated $D_{xj}$ values correlate with intensified anthropogenic pressures and degraded habitat integrity. he habitat integrity index $Q_{xj}$ is computed as:

$$Q_{xj} = H_j \left( 1 - \left( \frac{D_{xj}^z}{D_{xj}^z + k^z} \right) \right)$$

(4)

The habitat integrity metric $Q_{xj}$ quantifies ecological conditions in grid cell $x$ of land cover class $j$, where $H_j$ is baseline habitat suitability, $D_{xj}$ is cumulative threat intensity (computed per InVEST protocol), z is scaling parameter (default 2.5), and k is saturation coefficient calibrated as 0.5·max $Dxj$.

Building upon regional context, five anthropogenic stressors were operationalized: croplands, construction zones, rural dwellings, barren lands, and transportation corridors. Parameterization of threat decay distances and relative weights followed InVEST protocols and empirical validations [29,43,44] (Tables 1–2).

**2.3.2 Kernel Density Estimation (KDE).** Kernel density estimation (KDE) is mainly used to analyze the degree of equilibrium of density within the perimeter of an element, with higher values of kernel density indicating a denser distribution of tourist sites [45]. In this study, KDE was applied to visualize the spatial pattern of tourism POIs in ArcGIS 10.8 based on the acquired tourism POI data [46]. Analyzing the degree of aggregation of tourist site elements within the perimeter of the raster image element by setting the appropriate bandwidth. In calculating the density of scenic spots, taking into account the different influences of scenic spots, the scenic spots are assigned a value of 1–5 according to the scenic spot level. the search radius of scenic spots and hotels is uniformly set at 10 km, and the search radius of other elements is set at 15 km. The fuzzy membership tool in Arcgis10.8 was used to normalize the kernel density raster data, and the spatial distribution map of the nuclear density of tourism elements was obtained by superposition. The specific calculation formula is as follows:

$$f_n(x) = \frac{1}{nh} \sum_{i=1}^{n} K \left( \frac{x - x_i}{h} \right)$$

(5)

Where: $K \left( \frac{x-x_i}{h} \right)$ is kernel weighting function; $h$ is smoothing parameter (bandwidth); $x - x_i$ is Euclidean distance between estimation point $x$ and observed point $x_i$.

**2.3.3. Quantification of tourism activity intensity and zoning.** The intensity of tourism activities refers to the degree of impact of tourism activities on the natural environment, human resources and regional economic and social development, and the intensity of tourism activities can be characterized by the intensity of tourism elements [47]. The six elements of tourism include food, accommodation, transportation, tourism, shopping and entertainment, of which scenic spots, accommodation, food and beverage, shopping and leisure and entertainment facilities are selected as elements for analysis. Transportation, as infrastructure, covers a wide range and is not included for the time being. Through the kernel

**Table 1. Weight assignment and maximum impact distance of threat factors.**

| Threat | Max_ dist/ km | Weight | Decay |
|---|---|---|---|
| Cropland | 2.5 | 0.4 | Linear |
| Townland | 4 | 1 | Exponential |
| Rural settlements | 3 | 0.8 | Exponential |
| Other construction land | 5 | 0.9 | Linear |
| Unused land | 2 | 0.5 | Linear |
| Roads | 3 | 0.6 | Exponential |

**Table 2. Sensitivity of land types to threat factors.**

| Land use type | Habitat suitability | Cropland | Townland | Rural Settlements | Other construction land | Unused Land | Roads |
|---|---|---|---|---|---|---|---|
| Paddy Fields | 0.6 | 0.3 | 0.5 | 0.4 | 0.4 | 0.1 | 0.6 |
| Dry Farmland | 0.4 | 0.3 | 0.5 | 0.4 | 0.4 | 0.1 | 0.5 |
| Dense Forest | 1 | 0.8 | 1 | 0.8 | 0.6 | 0.2 | 0.8 |
| Shrubland | 1 | 0.4 | 0.6 | 0.5 | 0.5 | 0.1 | 0.7 |
| Sparse Forest | 1 | 0.9 | 1 | 0.9 | 0.6 | 0.3 | 0.7 |
| Other Woodland | 1 | 0.9 | 1 | 0.9 | 0.6 | 0.3 | 0.7 |
| High-coverage Grassland | 0.8 | 0.4 | 0.6 | 0.4 | 0.5 | 0.2 | 0.65 |
| Medium-coverage Grassland | 0.7 | 0.5 | 0.7 | 0.5 | 0.6 | 0.3 | 0.6 |
| Low-coverage Grassland | 0.6 | 0.5 | 0.6 | 0.6 | 0.6 | 0.3 | 0.6 |
| Rivers/Canals | 1 | 0.7 | 0.9 | 0.8 | 0.8 | 0.1 | 0.5 |
| Lakes | 1 | 0.7 | 0.9 | 0.8 | 0.8 | 0.2 | 0.5 |
| Reservoirs/Ponds | 1 | 0.7 | 0.9 | 0.8 | 0.8 | 0.2 | 0.5 |
| Permanent Glacier/Snow | 1 | 0.5 | 0.1 | 0.1 | 0.7 | 0.2 | 0 |
| Beach Land | 0.6 | 0.7 | 0.8 | 0.7 | 0.7 | 0.3 | 0.2 |
| Urban Land | 0 | 0 | 0 | 0 | 0 | 0 | 0 |
| Rural Settlements | 0 | 0 | 0 | 0 | 0 | 0 | 0 |
| Other Built-up | 0 | 0 | 0 | 0 | 0 | 0 | 0 |
| Sandy Land | 0 | 0 | 0 | 0 | 0 | 0 | 0 |
| Gobi Desert | 0 | 0 | 0 | 0 | 0 | 0 | 0 |
| Marshland | 0.5 | 0.4 | 0.4 | 0.2 | 0.3 | 0.1 | 0.3 |
| Bare Soil | 0 | 0 | 0 | 0 | 0 | 0 | 0 |
| Rocky Land | 0 | 0 | 0 | 0 | 0 | 0 | 0 |
| Others | 0 | 0 | 0 | 0 | 0 | 0 | 0 |

density superposition analysis of the five elements, the intensity of tourism activities in the region is divided, and the intensity of tourism activities in the final raster can be expressed as:

$$f_T = f'_A + f'_H + f'_R + f'_S + f'_E \tag{6}$$

In the equation: $f_T$ is the density of tourism activities; $f'_A$ is the density of tourist attractions after normalization; $f'_H$ is the density of accommodation facilities after normalization; $f'_R$ is the density of catering enterprises after normalization; $f'_S$ is the density of shopping places after normalization; $f'_E$ is the density of leisure and entertainment places after normalization.

**2.3.4. GeoDetector analysis.** The GeoDetector technique was implemented to disentangle drivers of habitat quality spatial heterogeneity within tourism pressure gradients. Twelve predictors were analyzed: five biophysical variables (X1: Average annual temperature, X2: Average annual precipitation, X3: Elevation, X4: Slope, X5: Aspect) and seven anthropogenic indicators (X6: Tourists, X7: GDP, X8: population density, X9: Lodging Density X10: Catering density, X11: Attraction density, X12: Road Network distribution density.). Specific statistics were given in the supplementary material (S1 Table). The explanatory force ($q$ value) of each factor for the spatial differentiation of habitat mass by factor detectors, and the interaction type between factors is analyzed using the interactive detector. It is divided into independent, two-factor enhancement, nonlinear enhancement, nonlinear attenuation and single-factor nonlinear attenuation (S2 Table). The calculation formula is as follows [48,49]:

$$q = 1 - \frac{\sum_{h=1}^{L} n_h \sigma_h^2}{n\sigma^2} \tag{7}$$

The equation parameters are defined as: $q$ is degree of interpretation of the driver factor; $n_h$ is sample size of the h-th layer driver factor; $\sigma_h$ is variance of the h-th layer driver factor; n is total sample size; $\sigma^2$ is total variance. The $q$ statistic ($0 \leq q \leq 1$), the larger the value, the higher the explanation.

## 3 Results

### 3.1 Spatial distribution characteristics of habitat quality

**3.1.1 Prefecture-scale habitat quality zonation.** Employing ArcGIS's Equal Interval classification, habitat integrity across Gannan was categorized into five distinct classes (Fig 2), with detailed areal distributions quantified per class (Table 3).

The area and proportion of habitat quality of different levels in GTAP counties were counted (Table 4). High habitat zones (0.8−1) encompass 9457.71 km² (25.8% of total), predominantly located in Xiahe, Zhuoni, and Diebu counties, where Diebu County exhibits peak representation at 24.55%. Higher quality habitats (0.6–0.8) represent the most extensive category (33.44%), predominantly clustered in Maqu County (6959.86 km²) and Xiahe County (3940.27 km²), accounting for 33.44% and 18.93% respectively. Medium quality zones (0.4–0.6) demonstrate sparse spatial distribution patterns, where Xiahe County maintains the most substantial coverage (376.46 km², 21.19%), followed by Zhuoni County at 17.38%. Lower habitat sectors (0.2–0.4) display equitable areal distribution across the prefecture, with Maqu County dominating at 29.33% coverage. Low habitat zones (0–0.2) show maximal aggregation in Maqu County (692.96 km², 48.49%), with secondary concentration in Diebu County (19.37%).

The spatial configuration of habitat degradation across Gannan manifests integrated point-line-polygon patterns strongly associated with tourism infrastructure spatialization. Southwestern Maqu County displays clustered linear

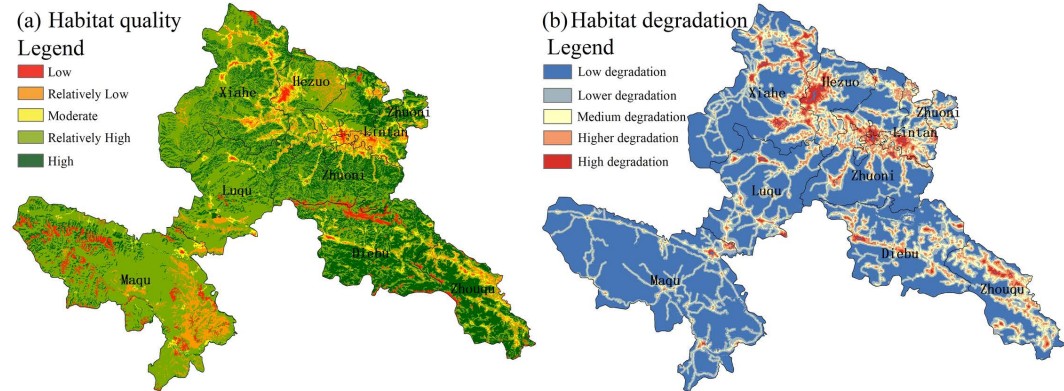

**Fig 2. Spatial distribution of habitat quality and habitat degradation in GTAP (base map and data from NGIC) Reprinted from NGIC under CC BY 4.0 license, with permission from "National Tibetan Plateau/ Third Pole Environment Data Center", original copyright 2017–2021.**

**Table 3. Statistics on the proportion of habitat quality grades in GTAP.**

| Habitat Quality Grade | Area (km²) | Percentage (%) |
| --- | --- | --- |
| Low [0–0.2] | 1429.18 | 3.9 |
| Relatively Low [0.2–0.4] | 3175.02 | 8.66 |
| Moderate [0.4–0.6] | 1776.68 | 4.85 |
| Relatively High [0.6–0.8] | 20812.52 | 56.79 |
| High [0.8–1] | 9457.71 | 25.8 |

**Table 4. Statistics on the proportion of habitat quality classification in cities and counties in GTAP.**

| County | Low [0-0.2] | | Relatively Low [0.2–0.4] | | Moderate [0.4-0.6] | | Relatively High [0.6–0.8] | | High [0.8-1] | |
|---|---|---|---|---|---|---|---|---|---|---|
| | Area (km²) | Percentage % | Area (km²) | Percentage % | Area (km²) | Percentage % | Area (km²) | Percentage % | Area (km²) | Percentage % |
| Hezuo | 71.89 | 5.03 | 401.45 | 12.64 | 211.28 | 11.89 | 1138.31 | 5.47 | 271.34 | 2.87 |
| Maqu | 692.96 | 48.49 | 931.19 | 29.33 | 178.61 | 10.05 | 6959.86 | 33.44 | 839.77 | 8.88 |
| Luqu | 36.36 | 2.54 | 221.61 | 6.98 | 126.79 | 7.14 | 3072.45 | 14.76 | 879.86 | 9.30 |
| Zhouqu | 99.57 | 6.97 | 357.49 | 11.26 | 195.95 | 11.03 | 844.12 | 4.06 | 1544.5 | 16.33 |
| Diebu | 276.8 | 19.37 | 141.26 | 4.45 | 170.91 | 9.62 | 1827.44 | 8.78 | 2321.51 | 24.55 |
| Lintan | 53.83 | 3.77 | 347.93 | 10.96 | 207.93 | 11.70 | 436.5 | 2.10 | 352.79 | 3.73 |
| Zhuoni | 116.95 | 8.18 | 335.97 | 10.58 | 308.75 | 17.38 | 2593.58 | 12.46 | 1796.71 | 19.00 |
| Xiahe | 80.83 | 5.66 | 438.12 | 13.80 | 376.46 | 21.19 | 3940.27 | 18.93 | 1451.23 | 15.34 |

degradation along transport corridors, contrasting with contiguous degradation zones in southeastern and northern sectors. High-intensity degradation clusters occur in urban tourism nuclei (Hezuo-Lintan corridor) and transportation corridors, reflecting concentrated anthropogenic pressures from tourism fluxes and land conversion. Secondary degradation belts follow roadway matrices, demonstrating infrastructure-mediated ecosystem perturbation. Meso-scale degradation characterizes peri-urban interfaces, whereas low-intensity degradation prevails in remote high-elevation territories experiencing limited anthropogenic intrusion.

Spatial analysis reveals pronounced ecological integrity in northeastern Zhuoni, southwestern Diebu, and Zhouqu counties, contrasting with degraded habitats in Hezuo City, Lintan, and Maqu counties, collectively forming an east-west biogeographic divergence across the prefecture.

**3.1.2 Township-scale habitat quality zonation.** As primary administrative divisions, townships provide an optimal resolution for deciphering habitat quality gradients in Gannan, enabling fine-scale analysis of biogeographic heterogeneity while informing precision ecological management. We aggregated township-level habitat metrics (Fig 3), demonstrating significant associations between ecological integrity indices and vegetation composition-spatial configuration patterns.

A distinct ecological gradient emerges across Gannan's townships, with habitat integrity declining progressively from alpine southeastern ranges to northwestern foothills and sedimentary basins. Superior ecological integrity zones (dark green) concentrate in southeastern highlands, particularly Diebu, Zhouqu, and Zhuoni counties, exemplified by Chagang and Gongba townships. Characterized by pristine montane ecosystems and dense vegetation canopies, these high-elevation territories constitute Gannan's ecological tourism heartland. Iconic protected landscapes such as Zagana and Lazikou maintain low-impact tourism thresholds coupled with rigorous conservation, sustaining peak habitat integrity indices across the prefecture. Secondary quality habitats (light green) occupy ecotones between natural systems and human-modified landscapes, notably Kecai Township in Xiahe's buffer zones. Balancing pastoral tourism and cultural preservation, these areas sustain moderate visitation levels that permit stable habitat functionality. Intermediate habitat states (yellow) characterize peri-urban interfaces where controlled rural tourism intersects with managed resource utilization. Degraded habitat corridors (orange/red) aggregate in western urbanization axes, particularly Hezuo's metropolitan sphere and Maqu's transportation hubs. While functioning as tourism growth poles through enhanced connectivity, these areas endure pronounced ecological fragmentation from land conversion and linear infrastructure proliferation. Cumulative anthropogenic pressures on vulnerable ecosystems have sculpted a biocapacity-depleted corridor along central-northern sectors, manifesting as sustained habitat degradation.

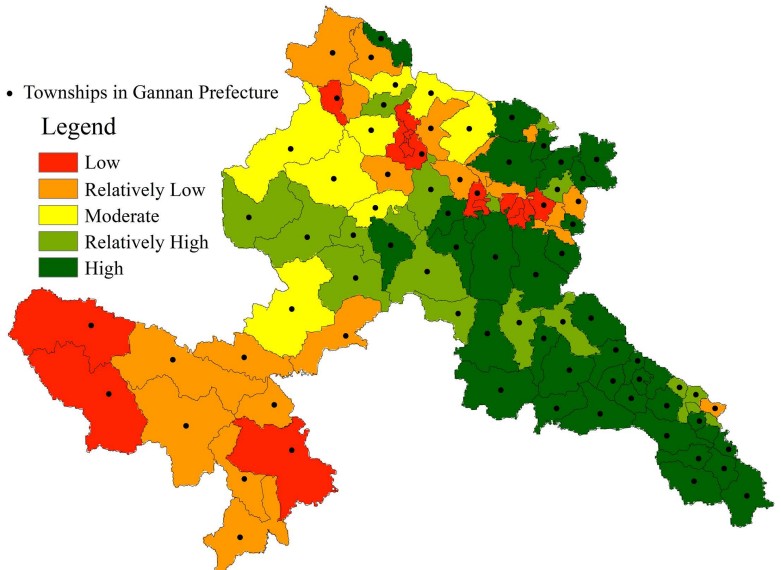

**Fig 3. Results of Township-Scale Habitat Quality (base map and data from NGIC) Reprinted from NGIC under CC BY 4.0 license, with permission from "National Tibetan Plateau/ Third Pole Environment Data Center", original copyright 2017–2021.**

### 3.2. Spatial characteristics of tourism activity intensity

While lacking conventional urban tourism infrastructure, Gannan's ascendance as a biocultural destination stems from synergistic growth of heritage tourism and nature-based recreation, driven by pristine ecosystems and Tibetan cultural capital. Tourism pressure hotspots manifest as spatial aggregations of infrastructure density, measurable through multivariate density indices of key service components. Our analysis operationalizes five pressure indicators: attractions, lodging, dining, retail, and leisure facilities, excluding transportation networks given their ubiquitous distribution as baseline infrastructure. Multilayer kernel density estimation (KDE) of tourism infrastructure enables spatial explicition of anthropogenic pressure gradients, creating empirical baselines for analyzing tourism-ecology coupling mechanisms.

1826 restaurants, 3776 shopping stores, 118 tourist attractions, 558 leisure and entertainment spots and 1132 accommodation facilities in GTAP were introduced into ArcGIS, and the 2023 GTAP tourism factor density map was generated (Fig 4). In calculating the density of the scenic area, considering the different influence of the scenic area, the scenic area was assigned 1–5 according to the scenic area level. The search radius of scenic spots and hotels is set to 10 km, and the search radius of other elements is set to 15 km.

Building upon the analytical framework, normalized Kernel Density Estimation (KDE) surfaces were integrated to produce composite tourism infrastructure density patterns (Fig 5). Employing Geometric Interval classification optimized for data distribution skewness, tourism pressure gradients were categorized as: peripheral tourism zones, transitional buffers, and tourism pressure epicenters [51], with quantitative spatial metrics detailed in Table 5.

Fig 5 reveals tourism pressure epicenters (11379.96 km², 31.04%) clustered around urban hubs and iconic sites (Zhagana, Labrang Monastery) in Hezuo-Zhuoni-Diebu corridors, demonstrating maximum service infrastructure density and visitation intensity. The largest spatial component (16444.63 km², 44.85%) comprises transitional buffers where anthropogenic pressures mediate cultural-natural landscape gradients through regulated tourism fluxes. Tourism wilderness areas (8840.84 km², 24.11%) preserve ecological integrity in peripheral regions, maintaining baseline biocultural conditions through minimal infrastructure penetration.

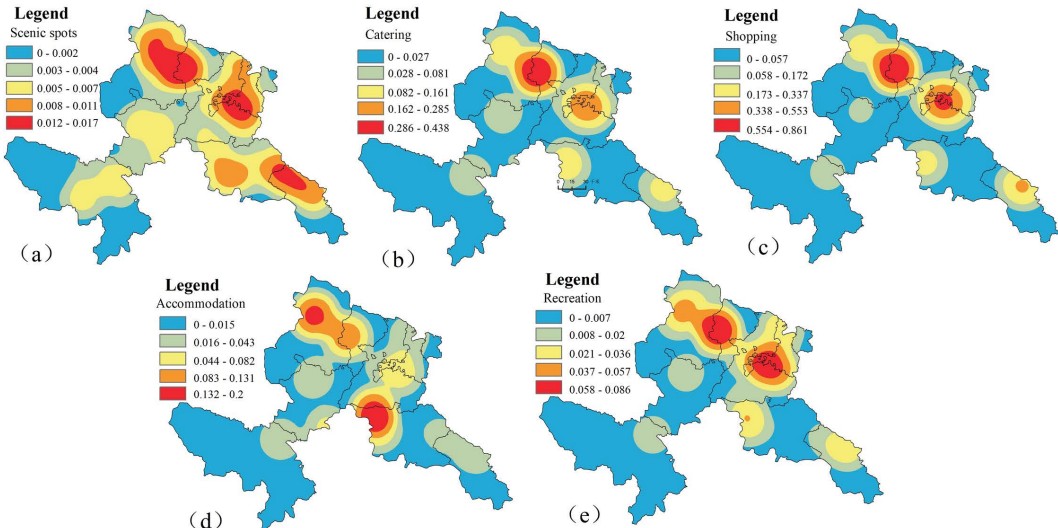

**Fig 4. Spatial distribution of the density of tourism core elements in GTAP in 2023 (base map and data from NGIC) (a, scenic spot core density; b, catering core density; c, shopping core density; d, accommodation core density; e, recreation core density).** Reprinted from NGIC under CC BY 4.0 license, with permission from "National Tibetan Plateau/ Third Pole Environment Data Center", original copyright 2017–2021.

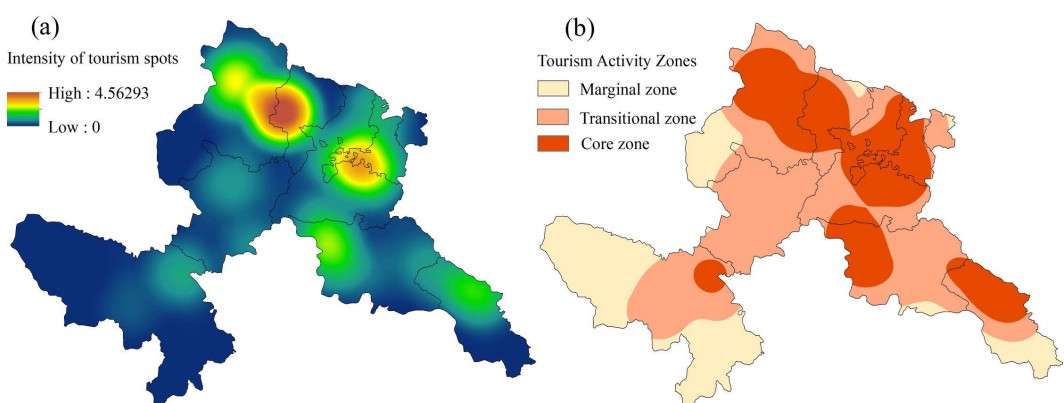

**Fig 5. Results of zoning of tourism activity intensity in GTAP (base map and data from NGIC) Reprinted from NGIC under CC BY 4.0 license, with permission from "National Tibetan Plateau/ Third Pole Environment Data Center", original copyright 2017–2021.**

**Table 5. GTAP Tourism Activity Intensity Zoning Information.**

| Tourism Activity Zone | Tourism Density Range | Area (km²) | Percentage% |
|---|---|---|---|
| Core Tourism Activity Zone | 0.875–4.56 | 11379.96 | 31.04% |
| Transitional Tourism Activity Zone | 0.144–0.87 | 16444.63 | 44.85% |
| Peripheral Tourism Activity Zone | 0–0.14 | 8840.84 | 24.11% |

A distinct tourism pressure cline emerges across Gannan's landscape, exhibiting gradational attenuation from service-dense cores to infrastructure-sparse peripheries, spatially coupled with tourism resource allocation patterns. Tourism pressure epicenters emerge through synergistic resource endowments and multimodal connectivity, contrasting with conservation-mediated activity suppression in ecological redlines across peripheral wilderness. The observed tourism

pressure stratification quantifies human-nature interface dynamics, providing empirical rationale for sustainable spatial zoning that balances visitation management with biodiversity sanctuary preservation.

### 3.3 Driving factors analysis of habitat quality spatial distribution

Post-tourism zoning stratification reveals distinct biophysical versus anthropogenic modulation of habitat heterogeneity patterns across partitioned landscapes. Quantified through q-statistics, variables exhibit tiered predictive capacity for habitat configuration variance along tourism pressure gradients. These results evidence scale-dependent interactions between ecological drivers and tourism pressures in shaping habitat dynamics across management regimes.

**3.3.1 Single factor analysis.** Table 6 shows the q value and ranking of habitat quality in different tourism intensity regions. Biophysical parameters emerge as principal determinants of habitat configuration heterogeneity, while tourism infrastructure demonstrates subordinate effects. A bioclimatic determinism gradient manifests spatially, where natural drivers' predictive capacity amplifies by 38% as tourism variables attenuate 62%, reflecting inverse dependencies on anthropogenic pressure clines.

A distinct core-periphery attenuation gradient manifests, exhibiting maximum intensity in tourism epicenters (core zones), moderate levels in transitional buffers, and minimal effects in ecological refugia (edge zones). Core zone analysis reveals tourism infrastructure as key pressure sources, with road networks ($q=0.051$) and Tourists ($q=0.049$) outperforming GDP effects ($q=0.083$), demonstrating hard infrastructure's dominance over economic metrics in habitat modification. Service facilities exhibit weak ecological perturbations (Catering: $q=0.036$; Lodging: $q=0.032$) despite high density, while attraction distribution shows minimal direct linkage ($q=0.011$), suggesting decoupling between tourism nodes and immediate habitat impacts. In transitional buffers, biophysical parameters regain control (Elevation: $q=0.143$; Slope: $q=0.142$), accounting for 58.7% of habitat variance, signaling landscape morphology's resurgence in mediating anthropogenic-natural interface dynamics. Residual tourism pressures persist (GDP: $q=0.117$; Tourists: $q=0.083$), demonstrating extended spatial of economic factors reach beyond immediate activity zones. Peripheral regions exhibit tourism variable collapse ($q<0.01$), enabling bioclimatic factors (Precipitation: $q=0.214$; Temperature: $q=0.192$) to reclaim 89.3% of habitat variance, confirming wilderness areas' resilience to anthropogenic pressures.

**3.3.2. Interaction detection.** Further analysis of interactive factors affecting habitat quality across core, transitional, and edge zones in Gannan reveals that natural factors (e.g., elevation X3, slope X4) exhibit stronger interaction effects, while tourism factors (e.g., attraction density X11, catering facility density X10, lodging density X9) demonstrate weaker

**Table 6. Detection of driving factors of habitat quality in different tourism intensity areas ($p<0.001$).**

| Factor | Core Tourism Zone | | Transitional Tourism Zone | | Peripheral Tourism Zone | |
|---|---|---|---|---|---|---|
| | q statistic | Rank | q statistic | Rank | q statistic | Rank |
| Annual Temperature X1 | 0.063 | 6 | 0.132 | 3 | 0.382 | 3 |
| Annual Precipitation X2 | 0.072 | 3 | 0.053 | 7 | 0.328 | 7 |
| Elevation X3 | 0.066 | 4 | 0.143 | 1 | 0.385 | 1 |
| Slope X4 | 0.065 | 5 | 0.142 | 2 | 0.384 | 2 |
| Aspect X5 | 0.084 | 1 | 0.107 | 5 | 0.364 | 5 |
| Tourists X6 | 0.049 | 9 | 0.083 | 6 | 0.338 | 6 |
| GDP X7 | 0.083 | 2 | 0.117 | 4 | 0.369 | 4 |
| Population Density X8 | 0.055 | 7 | 0.05 | 8 | 0.327 | 8 |
| Lodging Density X9 | 0.032 | 11 | 0.006 | 12 | 0.002 | 9 |
| Catering Density X10 | 0.036 | 10 | 0.009 | 10 | 0.001 | 10 |
| Attraction Density X11 | 0.011 | 12 | 0.041 | 9 | 0.00 | 12 |
| Road Network Density X12 | 0.051 | 8 | 0.008 | 11 | 0.00 | 11 |

synergistic impacts. At the prefecture scale (Fig 6a), the interaction between transportation network density (X12) and bioclimatic variables (X1 annual temperature, X3 elevation, X4 slope) exerts maximum influence on habitat quality, with an interaction $q$-value of 0.2. Overall, natural factor interactions show consistently higher $q$-values (mostly > 0.16), whereas tourism factor interactions display lower magnitudes, with minimum $q = 0.02$.

In core zones, natural-tourism factor interactions intensify significantly, with aspect (X5) × road density (X12) achieving $q = 0.17$, emerging as key habitat quality determinants. Tourism factor interactions escalate substantially (mean $q$ is close to 0.08), peaking at X7 GDP × X12 road density ($q = 0.14$), demonstrating non-negligible compound impacts on core zone habitats. Transitional and edge zones witness enhanced natural factor synergies alongside diminished tourism interactions. Edge zone tourism interactions (X9-X12) approach $q$ is close to 0, indicating negligible compound effects on habitat quality.

Cross-factor analysis confirms natural interactions dominate habitat spatial patterning, outperforming individual factors through predominant bifactorial enhancement mechanisms. This underscores nature's pivotal role in habitat maintenance, while significant tourism interactions in core zones reveal ecosystem vulnerability to development pressures.

## 4. Discussion

### 4.1. Major drivers of habitat quality

The results of this study show that the habitat quality of GTAP shows a spatial distribution pattern of "high in the east and low in the west" and decreasing from southeast to northwest. This result is consistent with Zhou's research results on the

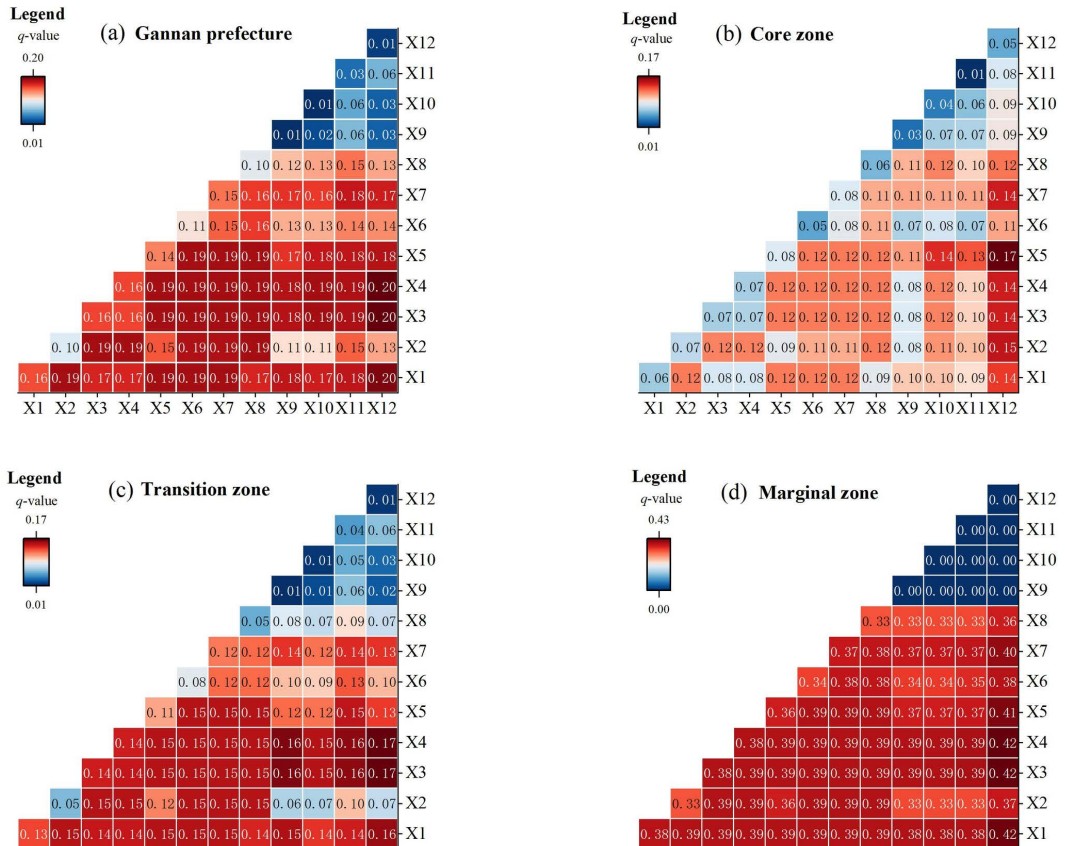

**Fig 6. Results of interaction between driving factors of habitat quality in different regions.** ($p < 0.001$).

habitat quality of GTAP [50]. Overall, the divergent influences on habitat quality in GTAP are dominated by natural factors, specifically related to the following factors: The low-altitude hilly mountainous areas in the southeastern Diebu County and Zhouqu County benefit from the East Asian monsoon and topographic uplift effect, with an average annual precipitation of 600–800 mm, average temperature is greater than 10°C, cumulative temperature of 1600–2000 d·°C, forming an advantageous area for water-heat coupling; The northwestern Hezuo city, Lintan County, etc. is located in the northeastern edge of the Tibetan Plateau transition zone, controlled by continental climate, precipitation is scarce and evaporation is strong, the ecological carrying capacity of the inherent disadvantage. In addition, the spatial distribution of habitat quality in GTAP is closely related to tourism activities. The ecotourism area in the southeast implements the mode of low interference development. In the urbanization area in the northwest, the expansion rate of construction land in Hezuo City is much higher than that in other areas. The high density of tourism traffic network leads to the fragmentation index of habitat obviously higher than that in the east. This is consistent with the research on the driving factors of habitat quality in tourist places by Wang [51], Peng Jian [52,53], Peng Liping [54] and others. The intensity of tourism activities in GTAP has a significant spatial correlation with habitat quality. The habitat quality characteristics of different tourism intensity areas reflect the impact of tourism factor aggregation on the ecological environment. In the core areas of high tourism factor density areas, habitat degradation is the most serious, the proportion of areas with low habitat quality has increased significantly, while the proportion of areas with high habitat quality has decreased. This shows that the aggregation of tourism factors is in a reverse relationship with the increase in habitat quality. With the gathering of tourism factors such as tourists, hotels and catering enterprises, the regional habitat quality will show a trend of accelerated degradation.

### 4.2. Relationship between habitat degradation and tourism activities

Habitat degradation degree indicates the degree of habitat affected by threat factors, and the higher the value, the more serious the degradation. The study found that the habitat degradation in GTAP showed the spatial distribution characteristics of "point, line and patch combination". Strongly degraded areas are concentrated in tourism core areas with high urbanization degree such as Hezuo City, Lintan County and Diebu County, river valleys and along traffic trunk lines. This is due to the fact that the more populated and urbanized towns in GTAP are concentrated in the ecologically better river valleys, and the concentration of human activities has led to significant habitat degradation in this area. It is worth mentioning that the distribution pattern of habitat degradation is consistent with the nuclear density results of tourism elements. We believes that changes in land use caused by tourism and human activities are the decisive factors of habitat degradation in GTAP. In addition, the interaction between traffic density (X12) and slope (X4) ($q = 0.28$) resulted in habitat degradation rate several times higher than that of pure nature in the transition zone of altitude 2000–3000 m, which indicated that road construction broke through the barrier effect of terrain and made tourism disturbance penetrate in ecological sensitive areas, forming a vicious cycle of "low natural resistance-high disturbance", especially in the west. The spatial aggregation of tourism elements leads to differences in tourism intensity, which affects habitat quality by affecting land development concentration and management effects. On the one hand, moderate intensity may slow down habitat degradation, but on the other hand, excessive intensity will aggravate degradation and change the distribution of habitat quality ratings.

### 4.3 Balance the tourism development and habitat conservation of GTAP

As a typical ecologically fragile plateau tourist destination, GTAP is almost prohibited from developing industries due to its special geographical location. In recent years, tourism has gradually become the dominant industry in the regional economic structure. How to achieve the "empowerment" of tourism to the development of ecologically fragile plateau areas is a key issue in the sustainable development of GTAP and even similar regions. A high-quality habitat is not only an important guarantee for the development of green economy, but also the core foundation for the sustainable development of plateau ecological tourism. Based on the results of this study, the following suggestions are made:

 

(1) The construction of a zoning control system: The core area needs to establish a threshold control on the intensity of tourism activities, strictly control the development density of scenic spots, and focus on restoring degraded habitats around scenic spots; Transitional areas should promote ecological tourism transformation, developing low-intensity experience projects; Fringe areas need to strengthen the principle of protection priority. It is recommended to include areas with altitude above 3800 m into the ecological red line, restrict the construction of roads and other infrastructure, and maintain the existing high-quality habitat coverage rate of 72.3%.

(2) Multi-factor collaborative management: The interaction effect of nature-tourism factors revealed by geodetector suggests that the "terrain-ecology-economic" comprehensive evaluation matrix needs to be established in tourism planning. Build a habitat quality warning system, when the growth rate of regional tourism revenue exceeds the growth rate of GDP by 15 percentage points, the ecological compensation mechanism will be triggered; It is recommended to pilot the "altitude gradient development" model in key areas of habitat degradation such as Hezuo cities.

(3) Dynamic monitoring system innovation: Transform the "InVEST-Kernel Density-Geodetector" technical framework into management tools. Develop a habitat quality monitoring platform in GTAP to achieve habitat degradation warning; Through drone remote sensing, ecological monitoring is implemented in the core area to ensure the qualification rate for remediation of degraded plaques.

### 4.3. Limitations and future research directions

This study has several limitations that warrant attention. First, while the InVEST model was employed to assess habitat quality, its parameter settings relied heavily on prior studies, potentially introducing subjective biases. Future research should incorporate field surveys and multi-source monitoring data to optimize key parameters based on local ecological conditions in GTAP, thereby enhancing assessment accuracy. Second, the current tourism activity analysis using POI (points of interest) data primarily reflects the spatial distribution of facilities (e.g., hotels, restaurants) but fails to capture their operational dynamics, such as occupancy rates or functional status (e.g., abandoned or underutilized sites). This limitation may skew the evaluation of tourism intensity; thus, integrating operational data (e.g., visitor foot traffic, facility utilization rates) is critical for more precise impact assessments. Third, the spatial analysis was confined to cross-sectional data from 2023, limiting insights into temporal variations. Longitudinal studies are urgently needed to unravel the dynamic interplay between ecological conservation and tourism development in highland regions and to establish adaptive management frameworks. Addressing these gaps will strengthen the scientific foundation for balancing ecosystem protection and sustainable tourism in Gannan and analogous fragile alpine ecosystems.

### 5. Conclusion

This study integrates GIS-based methods, including the InVEST model, kernel density analysis, and geodetectors, to assess habitat quality patterns, quantify tourism intensity, and delineate zones of tourism activity at multiple scales in GTAP [55]. The results of the study showed that the habitat quality in GTAP showed an obvious southeast-northwest decreasing gradient under the combined effect of natural conditions and anthropogenic pressure. High-quality habitats were mainly concentrated in Zhouqu, Diebu and Zhuoni counties in the southeast, which have superior topography and standardized eco-tourism; The degraded areas are mainly concentrated in the urbanized areas in the north-western part of the country (Hezuo City and Lintan County), showing a degradation pattern of "point - line - patch", mainly around the core tourist areas and transportation corridors. The kernel density analysis divides the GTAP into three zones of tourism intensity: the core zone (31.04% coverage, including the main towns and scenic areas), the transition zone (44.85%, a blend of cultural and natural landscapes), and the fringe zone (24.11%, the least disturbed and remote areas). Differences in habitat quality in these areas highlight the increasing impact of tourism on the ecosystem, with the core areas showing

the most severe degradation. Geodetector analysis showed that slope and elevation were the main natural drivers, while tourism factors exerted significant pressure on the core areas. Importantly, the interaction effect between natural and tourism factors amplified spatial differences.

This case study confirms that the sustainable development of tourism on the Tibetan Plateau must follow the synergistic logic of "ecological baseline constraints; development intensity adaptation; community benefit sharing". The proposed framework of "multi-scale diagnosis; multi-factor coupling; and multi-stakeholder collaboration" provides a scientific paradigm for coordinating tourism-driven economic and ecological restoration in the global plateau region. Future research should prioritize identifying the dynamic balance between the resilience of alpine ecosystems and the cycle of tourism disturbance, and ultimately promote the co-development of human activities and highland ecosystems.

## Supporting information

**S1 Table. Statistical data.**
(XLSX)

**S2 Table. Result types of two-factor interaction.**
(DOCX)

## Author contributions

**Conceptualization:** Bo Yang.

**Data curation:** Anna Miao.

**Validation:** Rong Jia.

**Visualization:** Jinlu Ran.

**Writing – original draft:** Jiawei Wu.

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
