## [Decision Letter · Decision Letter 0]

PONE-D-25-14264Balancing Tourism Development and Habitat Conservation in Fragile Ecosystems: A Case Study of the Qinghai-Tibet PlateauPLOS ONE

Dear Dr. Yang,

Thank you for submitting your manuscript to PLOS ONE. After careful consideration, we feel that it has merit but does not fully meet PLOS ONE’s publication criteria as it currently stands. Therefore, we invite you to submit a revised version of the manuscript that addresses the points raised during the review process.

**ACADEMIC EDITOR: **

According to the opinions of the two reviewers, please make the following modifications:

The introduction is rather confusing.  It should include the research background, literature review, and the author's research purpose.  At present, the literature review needs to be expanded.  A large part of it reviews previous studies on the Qinghai-Tibet Plateau.  However, in the research on plateaus, the Qinghai-Tibet Plateau may be one of them, and it can be further expanded from this type.  The research and analysis methods used by the author can also summarize and generalize the research of past scholars.  The research purpose should emphasize the author's innovation and contribution.The literature review should be re-organized to summarize the advances in this topic and present the shortages of existing relevant studies.The steps of data processing can be further clarified.  Currently, there is only a list of data sources.It should be noted that the current GIS visualization results are actually mainly aimed at two-dimensional plane content, and the specific problems of the ecosystem should have complex mechanisms and motivations behind them.  Can the author further analyze and summarize?It is necessary to add the relevant references to the manuscript, for example, the description of KDE.How to apply the findings of this study for practical applications?  The authors are suggested to add some discussions regarding the practical applications of the findings of this study.There are several errors/missing information in the reference section.  Pls carefully check.

We look forward to receiving your revised manuscript.

Kind regards,

Wenbin Nie

Academic Editor

PLOS ONE

“This research was supported by the 2021 Young Teachers’ Scientific Research Ability Improvement Plan of Northwest Normal University, the Innovation Fund for Colleges and Universities of the Department of Education, Gansu Province, China (grant no. 2021B-079), 2025 Graduate Research “Scientific Research Funding” Project, Gansu Province, China, and 2025 Graduate Research “Innovation Star” Project, Gansu Province, China.”

4. We note that Figures 1, 2, 3, 4 and 5 in your submission contain [map/satellite] images which may be copyrighted. All PLOS content is published under the Creative Commons Attribution License (CC BY 4.0), which means that the manuscript, images, and Supporting Information files will be freely available online, and any third party is permitted to access, download, copy, distribute, and use these materials in any way, even commercially, with proper attribution. For these reasons, we cannot publish previously copyrighted maps or satellite images created using proprietary data, such as Google software (Google Maps, Street View, and Earth). For more information, see our copyright guidelines: http://journals.plos.org/plosone/s/licenses-and-copyright.

1. You may seek permission from the original copyright holder of Figures 1, 2, 3, 4 and 5 to publish the content specifically under the CC BY 4.0 license. 

Reviewers' comments:

Reviewer's Responses to Questions

**Comments to the Author**

1. Is the manuscript technically sound, and do the data support the conclusions?

Reviewer #1: Yes

Reviewer #2: Yes

2. Has the statistical analysis been performed appropriately and rigorously? 

Reviewer #1: Yes

Reviewer #2: Yes

3. Have the authors made all data underlying the findings in their manuscript fully available?

Reviewer #1: Yes

Reviewer #2: No

4. Is the manuscript presented in an intelligible fashion and written in standard English?

Reviewer #1: Yes

Reviewer #2: Yes

5. Review Comments to the Author

Reviewer #1: The study aims to balance tourism development and habitat conservation in fragile ecosystems with the use of a case study of the Qinghai-Tibet plateau. Overall, the study is interesting, and the manuscript is well-written. The reviewer has only a few comments.

1. The motivations and the contributions of this study are not presented clearly.

2. The literature review should be re-organized to summarize the advances in this topic and present the shortages of existing relevant studies.

3. It is necessary to add the relevant references to the manuscript, for example, the description of KDE.

4. How to apply the findings of this study for practical applications? The authors are suggested to add some discussions regarding the practical applications of the findings of this study.

5. There are several errors/missing information in the reference section. Pls carefully check.

Reviewer #2: (1) The introduction is rather confusing. It should include the research background, literature review, and the author's research purpose. At present, the literature review needs to be expanded. A large part of it reviews previous studies on the Qinghai-Tibet Plateau. However, in the research on plateaus, the Qinghai-Tibet Plateau may be one of them, and it can be further expanded from this type. The research and analysis methods used by the author can also summarize and generalize the research of past scholars. The research purpose should emphasize the author's innovation and contribution.

(2) The steps of data processing can be further clarified. Currently, there is only a list of data sources.

(3) It should be noted that the current GIS visualization results are actually mainly aimed at two-dimensional plane content, and the specific problems of the ecosystem should have complex mechanisms and motivations behind them. Can the author further analyze and summarize?

6. PLOS authors have the option to publish the peer review history of their article (what does this mean? ). If published, this will include your full peer review and any attached files.

**Do you want your identity to be public for this peer review?** For information about this choice, including consent withdrawal, please see our Privacy Policy .

Reviewer #1: No

Reviewer #2: **Yes: ** Yile Chen

---

## [Author Response · Author response to Decision Letter 1]

3 Jun 2025

Dear Academic Editors and Reviewers:

On behalf of all authors, I gratefully acknowledge your review comments for our manuscript “Balancing Tourism Development and Habitat Conservation in Fragile Ecosystems: A Case Study of the Qinghai-Tibet Plateau” (PONE-D-25-14264). We have revised the manuscript carefully in response to those comments, and in the rest of this letter, we have provided specific responses to each comment.

We hope that our replies and the resulting changes will be acceptable, and that the manuscript will now be suitable for publication, but we will be happy to work with you to resolve any remaining issues.

Thank you very much once again for your assistance coordinating the review of the paper. The authors look forward to hearing about your decision.

Sincerely,

Bo Yang

College of Tourism,

Hexi Corridor Research Institute,

Northwest Normal University

Lanzhou 730070

China

Tel.: +86 18810599522

E-mail: yangbo907@126.com

Response to Reviewer #1 Comments

Reviewer #1: The study aims to balance tourism development and habitat conservation in fragile ecosystems with the use of a case study of the Qinghai-Tibet plateau. Overall, the study is interesting, and the manuscript is well-written. The reviewer has only a few comments.

No.1 The motivations and the contributions of this study are not presented clearly.

Answer: Thank you very much for the suggestions. According to the review opinions, we systematically strengthened the motivation and contribution of the research.

Revised content:

Motivation:

1.We clearly stated in the first paragraph of the Abstract.

As a typical representative of global ecologically fragile areas and emerging tourism hotspots, the Qinghai-Tibet Plateau has important research value in its collaborative path between ecosystem protection and tourism development. (see lines 12-14)

2. Several new parts have been added to the introduction, the origin of this study:

However, the development of tourism is often accompanied by changes in land use, which is one of the most important driving forces for ecosystem services [3, 4]. Non-natural factors of land use change can, to varying degrees, have far-reaching impacts on environmental quality [5], water cycle [6], carbon emissions [7], and biodiversity [8] in the region. Ecologically fragile areas are often economically underdeveloped regions, and the impact of tourism activities on land use change is most prominent in ecologically fragile areas [9]. Managers usually attach importance to the economic benefits brought by tourism development and ignore its impact on the environment, resulting in great damage to the ecosystem after tourism development exceeds the threshold of natural recovery [10]. Therefore, how to balance the relationship between tourism development and ecological protection in fragile ecosystems in the ecologically fragile areas represented by the Tibetan Plateau is an issue that managers, planners and tourism developers should continue to explore. The relationship between these two is of great significance in realizing the harmonization of tourism development and ecological protection in ecologically fragile areas. (see lines 59-73)

Contribution: In the last part of the introduction, our research contribution is added:

We use the InVEST model as an evaluation tool, and combines the methods of KDE and geodetector to explore the relationship between tourism development and ecological protection in Gannan Prefecture, a typical tourist destination in the northeastern margin of the Qinghai-Tibet Plateau. Specifically, we aim to: (1) Reveal the spatial distribution characteristics of HQs at different scales in Gannan Prefecture; (2) Reveal the distribution density of tourism facilities and divide the intensity zone of tourism activity; (3) Explore the driving factors of habitat quality differences under different tourist activity intensity zoning. (4) Explore the relationship between differences in habitat quality and intensity of tourism activities and develop targeted recommendations. (see lines 146-158).

No.2 The literature review should be re-organized to summarize the advances in this topic and present the shortages of existing relevant studies.

Answer: Thanks for the suggestion, We reorganized the literature review section to sort out the latest results of existing research topics and point out the shortcomings of related research.

Revised content:

Literature review is in the middle of the introduction:

Habitat quality (HQ) is the ability of the natural environment to provide suitable conditions for the continued survival and development of individuals and populations, it is a reflection of the size of the natural environment’s ability to support the survival and reproduction of organisms, and a key indicator of ecosystem service function and biodiversity level [11]. The quality of habitat not only reflects the ecosystem service capacity and biodiversity level, but also directly affects ecological security and human well-being [12, 13]. In recent years, with the rapid development of computer technology, 3S (GPS, GIS, RS) technology and big data technology, a series of ecological models have been introduced for quantitative assessment of habitat quality, including the ecological niche model (ENM) [14], the habitat suitability index model (HSIM) [15], the social value model of ecosystem services (SolVES) [16], the multi-scale ecosystem service integration model (MIMES) [17], the artificial intelligence of ecosystem service model (ARIES) [18], the pressure-state-response model (PSR) [19], and the Integrated Valuation of Ecosystem Services and Tradeoffs model(InVEST) [20 - 22]. Among them, the InVEST model is widely used because of its ease of application, data availability, and strong theoretical foundation, which makes it possible to conduct studies at different scales [23, 24].

In related research on habitat quality, research on habitat quality assessment [25], analysis of spatiotemporal evolution characteristics [26 -28], driving factors [29] and influencing mechanism [30] have become key issues. For example, Parvar assessed spatial and temporal changes in HQ in the Iranian Plateau using the optimized InVEST model, and their findings highlighted that long-term climate change and urbanization processes have led to a continuous decline in HQ over the past 30 years, and that HQ is predicted to decrease further by 2050 [31]; Yohannes analyzed the main causes of degradation of HQ in the Ethiopian highlands from 1972-2017, noting that landscape composition was greater than watershed impacts [32]�Han used mathematical statistics combined with InVEST model to reveal the impact mechanism of HQ in the arid urban agglomeration in the ecologically fragile areas of the northern slope of Tianshan Mountain, explored the interaction between natural and unnatural factors, and established an ecological-economic influence path [33]. In recent years, the impact of tourism-based human activities on biodiversity and HQ in ecologically fragile areas has become a hot topic of research. Luo et al. discussed the impact of different tourism interference intensity on the habitat quality and population size of giant salamanders in the Zhangjiajie Nature Reserve in Hunan Province, and found that noise and pathogens brought by tourism can directly or indirectly reduce the size of giant salamanders populations [34]; Taking Yulong River Basin as an example, Sun et al. explored the coupling relationship between “tourism development-land use-landscape pattern” in ecologically fragile karst areas based on 3S technology and POI spatial analysis [35], it provides a new reference for the impact of the rapid development of tourism on land use change and ecological environment; Hu et al. used the InVEST habitat quality and KDE method, combined with the distribution density of ethnic minority village spaces, to determine the potential development space of QMDA tourism resources, emphasizing the delicate balance between tourism development and ecosystem protection [36]. However, relevant research on HQ in ecologically fragile zones mainly stays from the perspective of ecology and geography, and there seems to be very few research on exploring the connection between tourism development and ecological protection.

The Tibetan Plateau region, with its unique geomorphologic and climatic conditions, has become an ecological protection barrier in China and at the same time a typical ecologically fragile area [37]. In recent years, it has not only become a key area for HQ research, but also one of the most popular tourist destinations in China due to its unique ecological and cultural values [38, 39]. In 2024, the Tibetan Plateau region will receive 117 million tourists and realize a comprehensive tourism income of 126.659 billion yuan. As a typical interactive area of human-land relationship, the tourism land on the Tibetan Plateau combines high-quality natural and humanistic resources, but at the same time there is a contradiction between ecological protection and tourism development [40]. Currently, most of the research on tourism sites on the Tibetan Plateau focuses on exploring the development of tourism industry, tourism safety assessment, and tourism competitiveness enhancement. In contrast, there is still relatively little exploration of its HQ, and the relationship between its HQ and tourism development needs to be further clarified. Therefore, we selected Gannan Tibetan Autonomous Prefecture, a typical plateau tourist at the northeast edge of the Qinghai-Tibet Plateau, and deeply explored the spatial distribution characteristics and main influencing factors of its habitat quality, revealing the impact of tourism development on habitat quality in the Qinghai-Tibet Plateau area. The area has a unique plateau ecosystem and rich Tibetan cultural tourism resources. The rapid development of tourism has become one of the important forms of human activities. As of 2023, Gannan Prefecture has a total of 36 A-level scenic spots, receiving 22 million tourists throughout the year and achieving tourism revenue of 11 billion yuan. The tourism industry has become an important pillar of economic development. However, the rapid development of tourism activities and related infrastructure construction has exacerbated grassland degradation, forest reduction and wetland shrinkage, posing a threat to habitat quality and ecosystem stability [41]. (see lines 74-145)

No.3 It is necessary to add the relevant references to the manuscript, for example, the description of KDE.

Answer: Thanks for the suggestion, we have added relevant references of research methods to the manuscript. Related references have been added to other research methods (see lines 202-283), references [24], [29] [42] [43] [44] [47].

Revised content:

Revised descriptions of KDE: In this study:

Kernel density estimation (KDE) is mainly used to analyze the degree of equilibrium of density within the perimeter of an element, with higher values of kernel density indicating a denser distribution of tourist sites [45]. In this study, KDE was applied to visualize the spatial pattern of tourism POIs in ArcGIS 10.8 based on the acquired tourism POI data [46]. Analyzing the degree of aggregation of tourist site elements within the perimeter of the raster image element by setting the appropriate bandwidth. In calculating the density of scenic spots, taking into account the different influences of scenic spots, the scenic spots are assigned a value of 1 to 5 according to the scenic spot level. the search radius of scenic spots and hotels is uniformly set at 10km, and the search radius of other elements is set at 15km. The fuzzy membership tool in Arcgis10.8 was used to normalize the kernel density raster data, and the spatial distribution map of the nuclear density of tourism elements was obtained by superposition. (see lines 242-258, references [45] [46])

No.4 How to apply the findings of this study for practical applications? The authors are suggested to add some discussions regarding the practical applications of the findings of this study.

Answer: Thanks for the suggestion, We have added a new paragraph in the discussion section to illustrate the practical application of the results of this study.

Revised content:

4.3 Balance the Tourism Development and Habitat Conservation of GTAP

As a typical ecologically fragile plateau tourist destination, Gannan Prefecture is almost prohibited from developing industries due to its special geographical location. In recent years, tourism has gradually become the dominant industry in the regional economic structure. How to achieve the “empowerment” of tourism to the development of ecologically fragile plateau areas is a key issue in the sustainable development of GTAP and even similar regions. A high-quality habitat is not only an important guarantee for the development of green economy, but also the core foundation for the sustainable development of plateau ecological tourism. Based on the results of this study, the following suggestions are made:

(1) The construction of a zoning control system: The core area needs to establish a threshold control on the intensity of tourism activities, strictly control the development density of scenic spots, and focus on restoring degraded habitats around scenic spots; Transitional areas should promote ecological tourism transformation, developing low-intensity experience projects; Fringe areas need to strengthen the principle of protection priority. It is recommended to include areas with altitude above 3800 m into the ecological red line, restrict the construction of roads and other infrastructure, and maintain the existing high-quality habitat coverage rate of 72.3%.

(2) Multi-factor collaborative management: The interaction effect of nature-tourism factors revealed by geodetector suggests that the “terrain-ecology-economic” comprehensive evaluation matrix needs to be established in tourism planning. Build a habitat quality warning system, when the growth rate of regional tourism revenue exceeds the growth rate of GDP by 15 percentage points, the ecological compensation mechanism will be triggered; It is recommended to pilot the “altitude gradient development” model in key areas of habitat degradation such as Hezuo cities.

(3) Dynamic monitoring system innovation: Transform the “InVEST-Kernel Density-Geodetector” technical framework into management tools. Develop a habitat quality monitoring platform in GTAP to achieve habitat degradation warning; Through drone remote sensing, ecological monitoring is implemented in the core area to ensure the qualification rate for remediation of degraded plaques. (see lines 528-558)

No.5 There are several errors/missing information in the reference section. Pls carefully check.

Answer: Thanks for the suggestion, we have revised the references and modified them according to the format in the PLOS's submission guide.

Revised content:

References (see lines 606-795)

Response to Reviewer #2 Comments

No.1. The introduction is rather confusing. It should include the research background, literature review, and the author's research purpose. At present, the literature review needs to be expanded. A large part of it reviews previous studies on the Qinghai-Tibet Plateau. However, in the research on plateaus, the Qinghai-Tibet Plateau may be one of them, and it can be further expanded from this type. The research and analysis methods used by the author can also summarize and generalize the research of past scholars. The research purpose should emphasize the author's innovation and contribution.

Answer: Thanks for the suggestion. We have revised the introduction, and the new introduction mainly includes three parts: 1. Research background (first paragraph) 2. Expand the literature review on this topic, focusing on other ecologically fragile tourist destinations and other plateau areas in the world. Based on our research methods, the research and analysis methods of previous scholars have been summarized and summarized. 3. Emphasize the research purpose of this article (including innovation, motivation and contribution)

Revised content:

Research background (see lines 50-73)

According to the United Nations “World

---

## [Decision Letter · Decision Letter 1]

Balancing tourism development and habitat conservation in fragile ecosystems: A case study of the Qinghai-Tibet Plateau

PONE-D-25-14264R1

Dear Dr. Yang,

We’re pleased to inform you that your manuscript has been judged scientifically suitable for publication and will be formally accepted for publication once it meets all outstanding technical requirements.

Kind regards,

Wenbin Nie

Academic Editor

PLOS ONE

Additional Editor Comments (optional):

Reviewers' comments:

Reviewer's Responses to Questions

**Comments to the Author**

1. If the authors have adequately addressed your comments raised in a previous round of review and you feel that this manuscript is now acceptable for publication, you may indicate that here to bypass the “Comments to the Author” section, enter your conflict of interest statement in the “Confidential to Editor” section, and submit your "Accept" recommendation.

Reviewer #1: (No Response)

Reviewer #2: All comments have been addressed

2. Is the manuscript technically sound, and do the data support the conclusions?

Reviewer #1: (No Response)

Reviewer #2: Yes

3. Has the statistical analysis been performed appropriately and rigorously? 

Reviewer #1: (No Response)

Reviewer #2: Yes

4. Have the authors made all data underlying the findings in their manuscript fully available?

Reviewer #1: (No Response)

Reviewer #2: Yes

5. Is the manuscript presented in an intelligible fashion and written in standard English?

Reviewer #1: (No Response)

Reviewer #2: Yes

6. Review Comments to the Author

Reviewer #1: (No Response)

Reviewer #2: All comments have been addressed. The author presents the revisions in a clear responsive manner, and I think the paper is now well organized, and I recommend it for publication.

7. PLOS authors have the option to publish the peer review history of their article (what does this mean? ). If published, this will include your full peer review and any attached files.

**Do you want your identity to be public for this peer review?** For information about this choice, including consent withdrawal, please see our Privacy Policy .

Reviewer #1: No

Reviewer #2: **Yes: ** Yile Chen

---

## [Editor Report · Acceptance letter]

PONE-D-25-14264R1

PLOS ONE

Dear Dr. Yang,

I'm pleased to inform you that your manuscript has been deemed suitable for publication in PLOS ONE. Congratulations! Your manuscript is now being handed over to our production team.

Kind regards,

on behalf of

Dr. Wenbin Nie

Academic Editor

PLOS ONE